# Detection of Tick-Borne Pathogens in Red Deer (*Cervus elaphus*), United Kingdom

**DOI:** 10.3390/pathogens10060640

**Published:** 2021-05-23

**Authors:** Nicholas Johnson, Megan Golding, Laurence Paul Phipps

**Affiliations:** Arbovirus Research Team, Virology Department, Animal and Plant Health Agency (Weybridge), Woodham Lane, Addlestone KT15 3NB, UK; Megan.Golding@apha.gov.uk (M.G.); Paul.Phipps@apha.gov.uk (L.P.P.)

**Keywords:** *Babesia*, *Anaplasma phagocytophilum*, red deer, 18S rRNA gene, flavivirus

## Abstract

Deer represent a major vertebrate host for all feeding stages of the hard tick *Ixodes ricinus* in the United Kingdom (UK), and could play a role in the persistence of tick-borne pathogens. However, there have been few studies reporting the presence of *Babesia* spp. and *Anaplasma phagocytophilum* in deer in the UK, and those that detected *Babesia* were unable to confirm the species. To address this, we have investigated blood samples from red deer (*Cervus elaphus*) for the presence of tick-borne pathogens. Total DNA was extracted from haemolysed blood that was removed from clotted blood sampled from culled, captive red deer. *Babesia* spp. were detected with a pan-piroplasm PCR that amplifies a fragment of the 18S rRNA gene. Species were identified based on identity with published sequences. *Anaplasma phagocytophilum* was detected with a probe-based PCR targeting the *msp2* gene. In addition, residual serum samples from a subset of animals were tested for the presence of anti-flavivirus antibodies. Of 105 red deer samples tested from three locations in the United Kingdom, 5 were positive for piroplasm and 5 were positive for *A. phagocytophilum*. Co-infection with both pathogens was detected in two samples from one location. No evidence for antibodies against West Nile virus were detected. However, 12% of sera tested were positive for tick-borne encephalitis virus antibodies.

## 1. Introduction

Deer play an active role in the epidemiology of a range of livestock and human pathogens [1,2]. In the British Isles there are six species of deer. Two species are considered native, red deer (*Cervus elaphus*) and roe deer (*Capreolus capreolus*). A third species, fallow deer (*Dama dama*), was introduced during the medieval period. Three other species have established in the United Kingdom since their original introductions in the 19th century, Japanese sika (*Cervus nippon*), Reeve’s muntjac (*Muntiacus reevesi*) and Chinese water deer (*Hydropotes inermis*). Surveys in recent decades have suggested that they are all increasing in abundance and increasing in range, particularly the recently introduced species [3]. All are targets for tick predation as they generally occupy habitats where *Ixodes ricinus* ticks are prevalent, such as deciduous woodland and moorland. Based on these factors, deer have the potential to act as a key vertebrate reservoir for blood-borne pathogens.

The order Piroplasmida is dominated by two genera of tick-borne protozoal parasites, *Babesia* and *Theileria* [4]. In the United Kingdom, clinical disease caused by *Theileria* spp. is rarely encountered [5], but infection with *Babesia divergens* causes the main tick-borne disease of cattle, bovine Babesiosis, which is often termed redwater fever [6] and is found throughout the British Isles [7]. *Babesia divergens* is transmitted by *I. ricinus* and has been detected in cattle repeatedly over the past 100 years [8,9]. Other Babesias detected in ruminants include *B. major,* also in cattle [10], and *B. motasi* in sheep [11]. However, outbreaks of disease are sporadic and the maintenance of *Babesia* spp. within the tick population is likely to depend on wildlife species acting as the vertebrate host. *Anaplasma phagocytophilum* is a tick-borne Gram-negative bacterium in the family *Anaplasmataceae,* which has a worldwide distribution. It primarily infects neutrophils and, to a lesser extent, eosinophils, during the initial phase of the disease in vertebrate hosts, later infecting host lymphocytes. *Anaplasma phagocytophilum* is the etiological agent of human granulocytic anaplasmosis (HGA) in North America [12], but cases are rare in Europe [13]. By contrast, *A. phagocytophilum* infection is common in European livestock, causing tick-borne fever in cattle and sheep, as well as equine granulocytic anaplasmosis*. Anaplasma phagocytophilum* has been detected in British roe deer [14] and numerous studies from mainland Europe have detected the infection in roe deer [15,16], red deer [17,18] and fallow deer [19]. Co-infection with *Babesia* spp. is often reported.

The first evidence that UK deer could be involved in the epidemiology of bovine babesiosis was reported in the 1970s. Preliminary studies had reported that sera from Scottish red deer were seropositive for *B. divergens* [20]. This was followed by the isolation of *Babesia* species from the blood of wild deer [21]. The morphology of this *Babesia* was indistinguishable from *B. divergens* and *B. capreoli*, originally described by Enigk and Freidhoff in 1962 [22]. In addition to these two species, others have been identified in mainland Europe. Sequence from spleen samples from both roe and red deer in Slovenia [23] identified two species, one with high sequence identity to *B. divergens* and a second sharing identity with a species designated EU1 [24]. This species was also detected in French roe deer [25] and is now named *B. venatorum*. This species, like *B. divergens*, but not other *Babesia* species, is zoonotic and has been detected in UK ticks [26]. It has recently been reported in blood samples taken from healthy sheep in Scotland [27]. A further species has been detected in European deer and appears to be closely related to a *Babesia* found in white-tailed deer (*Odocoilus virginianus*) in the Great Dismal Swamp of Virginia, United States of America. The US *Babesia* was tentatively named *B. odocoilei* by Emerson and Wright [28]. More recently this species has been associated with deer from around the world [29,30,31] and is often referred to as *B. odocoilei*-like. A number of studies have surveyed for piroplasms in wild and farmed cervids within the Iberian Peninsula. In contrast to North European countries, surveys of deer have detected *Theileria* spp., with the occasional detection of *B. divergens,* in roe deer [32]. The only *Babesia* species detected in farmed red deer from central and southern Spain was *B. pecorum* [33], whilst a survey of wildlife in Portugal that included red and fallow deer, and wild boar (*Sus scrofa*), only detected *Theileria* spp. [34]. This may reflect the different tick assemblages found throughout the region, with drier areas dominated by *Hyalomma* species rather than *I. ricinus*. This suggests that there are at least four species of *Babesia* commonly encountered when surveying deer in northern Europe. A further development has been the detection of seropositivity for tick-borne encephalitis virus (TBEV) in deer sera and the isolation of TBEV from ticks in areas of high seroprevalence in Great Britain [35].

The persistence of livestock disease caused by *B. divergens* and *A. phagocytophilum* within the British Isles [9,26,36], increasing knowledge of *Babesia* diversity in wild ungulates in mainland Europe [37], and the discovery of piroplasmids that rarely cause livestock disease [5,27] has prompted us to investigate the presence of *Babesia* spp. and *A. phagocytophilum* in red deer from three locations within Scotland and England. We have also tested serum samples for the presence of antibodies against the flaviviruses West Nile virus (WNV) and TBEV.

## 2. Results

Blood samples were obtained from culled red deer from three locations, as outlined in Table 1. *Anaplasma phagocytophilum* was detected from three samples from Norfolk, one sample each from Dumfries and Galloway, and Lancashire. Piroplasm DNA was detected in two samples each from Norfolk and Dumfries and Galloway, and in one sample from Lancashire. Dual infection with piroplasm and *A. phagocytophilum* was detected in two samples (N406 and N408) from Norfolk. The level of detection within the pan-piroplasm assay as judged by the PCR threshold value (Ct) was relatively high in the deer samples (>35) compared to equivalent results for cattle with Babesiosis, where the threshold values are usually in the range of 20 to 25.

Sequencing of the pan-piroplasm amplicon generated a 363-base pair (bp) sequence from four samples (N406, DG368, DG393, L801). Sample N408 produced a partial sequence that was not subsequently analysed. Alignment of these sequences with publicly available Babesia 18S rDNA sequences demonstrated that they shared >99.7% sequence identity with those derived from red deer sampled in the Czech Republic [37]. Alignment of a 56 bp fragment including the hypervariable region amplified by primers Piro-A and Piro-B of deer-associated 18S rDNA sequences (*B. divergens*, *B. capreoli*, *B. venatorum* and deer-clade *Babesia*) indicated that deer-clades’ sequences, including the sequences generated in this study, contain a single base deletion within this region (Figure 1A). The other species generate a 364 bp sequence using these primers. A neighbour-joining analysis (Figure 1B) of deer-associated Babesia spp. showed the clustering of the UK sequence with the Czech “deer-clade” sequences with >99.7% sequence identity (GenBank Accession number MG344776) and diverges from a *B. odocoilei* sequence (AY046577) from North America with 100% bootstrap support. This cluster is separate from the major clade containing *B. venatorum*, *B. divergens,* and *B. capreoli*.

Following the recent detection of TBEV in ticks sampled in the UK associated with high levels of seropositivity in deer for this virus, the samples were also screened for flavivirus antibodies. No WNV antibodies were detected in any of the deer samples tested (Table 2). However, 12% of samples tested positive for TBEV by ELISA, the majority from deer sampled in Dumfries and Galloway (Table 2).

## 3. Discussion

A total of 105 blood samples obtained from British deer were tested for the presence of three tick-borne pathogens. Although sampled from three geographically separate areas, we detected the presence of a single *Babesia* species in each herd. Alignment with published *Babesia* 18S rDNA sequences suggests that this species is a deer-associated *Babesia* that has variously been termed *Odocoilei*-like, deer-clade or CH1. In addition, five animals were shown to be infected with *A. phagocytophilum*. This pathogen has been reported in Reeve’s muntjac in Norfolk in a previous study [38]. In two samples from Norfolk, both pathogens were detected. This survey did not detect the presence of other deer-associated *Babesia* spp., *B. divergens*, *B. capreoli* or *B. venatorum*. A previous investigation of disease in a herd of reindeer (*Rangifer tarandus tarandus*) in Scotland reported that seven animals died with clinical signs suggestive of Babesiosis [39]. Intraerythrocytic parasites were observed in blood smears and analysis of the 18S rDNA sequence suggested infection with *B. divergens*. Subsequent analysis concluded that this was an example of an infection with *B. capreoli* rather than *B. divergens* [40]. However, this separation is based on a small number of base changes within the 18S rDNA sequence, so this observation should be considered tentative. A study surveying red deer in Ireland suggested both *B. divergens* and *B. odocoilei*-like species were detected based on the identity of a partial 18S rDNA sequence [41]. The survey reported here suggests that UK deer are also infected, presumably sub-clinically as all the animals were healthy when sampled, with a *B. odocoilei*-like species. The high Ct values observed also suggest a low level of parasitemia when compared to cattle with Babesiosis. Alternatively, the extraction of DNA from a haemolysed blood clot may be sub-optimal compared to EDTA-treated blood, giving lower nucleic acid yields.

The absence of WNV antibodies in the deer samples is not surprising as the virus has not been isolated in the United Kingdom and seropositivity in horses is associated with vaccination [42]. Twelve percent of serum samples, mainly from the Scottish county of Dumfries and Galloway, were positive for TBEV. The close antigenic homology between TBEV and the endemic flavivirus Louping ill virus means that cross-reactivity can occur. Serology methods such as ELISA alone are not able to distinguish between the viruses. The infection of Louping ill virus is mainly reported in sheep, although seropositivity has been reported in deer [43]. Tick-borne encephalitis virus has recently been detected in UK deer by serological means and confirmed by the isolation of the virus from *I. ricinus* ticks from locations that are close to the deer herd sampled from Norfolk [35]. The seropositive results reported here could provide further evidence of TBEV circulation in UK deer, but further surveillance would be needed to confirm this.

## 4. Materials and Methods

Blood samples were taken post-mortem from farmed red deer routinely culled from three locations: the county of Dumfries and Galloway in Scotland; the counties of Norfolk in the East of England and Lancashire in the North of England. The numbers and sex of the deer are shown in Table 1. The samples were allowed to clot and the serum was removed. Residual haemolysed blood around the clot was removed and two hundred microlitres was used from each sample to extract total DNA using the QIAamp DNeasy kit (QIAgen, Manchester, UK) following the manufacturer’s protocol.

Each DNA sample was tested for the presence of piroplasms and *A. phagocytophilum*. A two microlitre aliquot was tested for *A. phagocytophilum* using a real-time PCR that amplifies a 75 bp fragment of the *msp*2 gene following the protocol of [44]. Piroplasms were detect in a 5 microlitre aliquot using a pan-piro PCR (primers Piro-A and Piro-B) that amplifies a 400 bp fragment of the 18S rRNA gene. The primers and amplification conditions have been published previously [45]. A positive control DNA extracted from the blood of a cow infected with both pathogens was used throughout [46]. The size of the amplicons was confirmed by separation on a 1.5% agarose gel stained with Sybr^®^Safe (Thermo Fisher Scientific, Horsham, UK) and visualized with UV illumination. Gel pilot (50 bp) DNA marker (QIAgen) was used to estimate band sizes. The amplicons were sequenced using the Piro-A and Piro-B primers using ABI PRISM^®^ BigDye^®^ Terminator v3.1 Cycle Sequencing Kit (Applied Biosystems, Paisley, UK). Sequences were edited using Lasergene version 12.1 (DNASTAR, Madison, WI, USA). The species of Babesia was confirmed by showing >99% identity with publically available sequences (NCBI GenBank database) following a BLAST search. Neighbour-joining analysis was generated using MEGA 5.2 [47] and bootstrap values produced by 1000 pseudoreplicates within the program.

Residual serum samples were tested for antibodies to WNV and TBEV using commercially available enzyme-linked immunosorbent assay (ELISA) kits. These were the ID Screen West Nile Competition Multi-Species (IDvet, Grabels, France) and the Immunozym FSME IgG All Species (PROGEN, Heidelberg, Germany), for WNV and TBEV, respectively. For the TBEV ELISA, samples with a reading of >126 Vienna units/mL were considered seropositive.

## 5. Conclusions

The zoonotic potentials of deer-clade and *odocoilei*-like *Babesia* species are uncertain. Cases of human Babesiosis are rare in the UK and associated with infection with *B. divergens* [48]. However, in the original case, species identification relied on the morphology of intraerythrocytic parasites and serology, and was prior to the development of DNA sequencing, the dominant means of assigning *Babesia* to species. For the other *Babesia* associated with human infection, *B. venatorum*, no human cases have been detected in the UK [26,27] and cases are rare in Europe [49]. Further studies in all deer species from a wider geographical range are needed to fully understand the *Babesia* species diversity present within the UK and the role deer play in their maintenance within the tick population.

## Figures and Tables

**Figure 1 pathogens-10-00640-f001:**
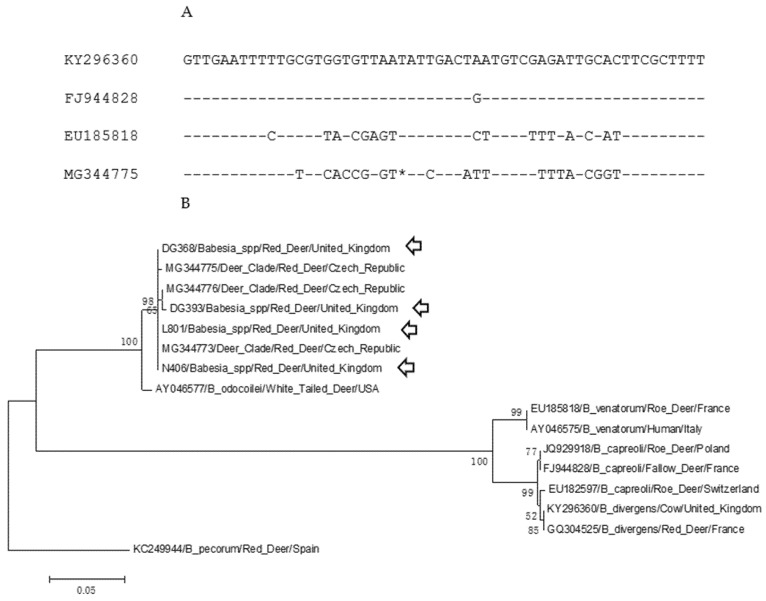
Sequence analysis of UK *Babesia* sequences with European ruminant-associated 18S rDNA sequences. (**A**) Alignment of the hypervariable regions of the 18S rDNA sequences between position 600 and 655. The sequences are KY296360 (*B. divergens*/cow/UK), FJ944828 (*B. capreoli*, fallow deer, France), EU185818 (*B. venatorum*/roe deer, France) and MG344775 (deer-clade *Babesia*, red deer, Czech Republic). The * indicates an insertion/deletion site within the hypervariable region that appears as a single nucleotide deletion within the deer-clade *Babesia*. (**B**) Neighbour-joining analysis of deer-associated *Babesia* spp. A 358 bp fragment of the 18S rDNA sequences was used for the analysis with 1000 bootstrap iterations. A *Babesia pecorum* sequence was used to root the phylogeny. Labels indicated the GenBank Accession number, *Babesia* spp., vertebrates host and country of origin. Arrows indicate sequences from the United Kingdom. Bootstrap values over 50 are shown.

**Table 1 pathogens-10-00640-t001:** Sample details of 105 red deer (*Cervus elaphus*) from three locations in the United Kingdom. Results for *A. phagocytophilum* and pan-piroplasm PCR on DNA extracted from each sample are given.

Location	Year/Samples	Total	*Anaplasma phagocytophilum* PCR (%) [Code Numbers]	Pan-Piroplasm spp. PCR (%) [Code Numbers]
Norfolk (N)	2020 0♂, 11♀	11	3/11 (27.3%) [N405, N406, N408]	2/11 (18.2%) [N406, N408]
Dumfries and Galloway (DG)	2020 50♂, 18♀	68	1/68 (1.5%) [DG422]	2/68 (2.9%) [DG368, DG393]
Lancashire (L)	2020 13♂, 13♀	26	1/26 (3.8%) [L811]	1/26 (3.8%) [L801]
Totals		105	5/105 (4.8%)	5/105 (4.8%)

**Table 2 pathogens-10-00640-t002:** Serosurvey for flaviviruses in deer samples from the United Kingdom.

Location	West Nile Virus ^1^ (ID Screen^®^ West Nile Competition Multi-Species) (%)	Tick-Borne Encephalitis Virus ^2^ (Immunozym FSME IgG All Species) (%)
Norfolk	0/9 (0)	0/9 (0)
Dumfries and Galloway	0/14 (0)	4/15 (26.7)
Lancashire	0/16 (0)	1/17 (5.9)
Total	0/39 (0)	5/41 (12.1)

^1^ positive values: S/N% < 40%; ^2^ positive values: >126 VIEU/mL. ^®^registered trademark

## Data Availability

Representative sequences from this study have been submitted to the GenBank under the accession numbers MW960417 for sample DG393 and MW969626 for sample DG368.

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
