# Peer review of "Detection of Tick-Borne Pathogens in Red Deer (Cervus elaphus), United Kingdom"

_pathogens, 2021, doi:10.3390/pathogens10060640_

Round 1

Reviewer 1 Report

The manuscript describes the detection of tick-borne pathogens in Red Deer in the United Kingdom.  Blood, collected from captive culled Red Deer was used for DNA isolation and tested by PCR for the presence of DNA from Babesia spp. and Anaplasma phagocytophilum.  A total of 105 samples from three locations in the UK were tested, and respective pathogens were detected in 5 out of 105  samples (4.8%).  Samples also were tested in virus neutralization assay for the presence of antibodies to West Nile virus (none were detected), and to tick-borne encephalitis virus (12% seroprevalence).

The manuscript is well written, experiment design and methodology are well described, and results are well discussed in the context of the relevant published literature.  The information on these pathogens in Red Deer in the UK is limited, so the manuscript will be of interest to the researchers in this area. 

There are only a few minor comments:   

Lines 157-158: “The low CT values observed also suggest a low level of parasitemia...”  Shouldn’t it be “ The high CT values…”?   

Line 178:  “…to extract total DNA using the QIAamp RNeasy kit…” Is this correct?

Author Response

The manuscript describes the detection of tick-borne pathogens in Red Deer in the United Kingdom.  Blood, collected from captive culled Red Deer was used for DNA isolation and tested by PCR for the presence of DNA from Babesia spp. and Anaplasma phagocytophilum.  A total of 105 samples from three locations in the UK were tested, and respective pathogens were detected in 5 out of 105  samples (4.8%).  Samples also were tested in virus neutralization assay for the presence of antibodies to West Nile virus (none were detected), and to tick-borne encephalitis virus (12% seroprevalence).

The manuscript is well written, experiment design and methodology are well described, and results are well discussed in the context of the relevant published literature.  The information on these pathogens in Red Deer in the UK is limited, so the manuscript will be of interest to the researchers in this area. 

There are only a few minor comments:   

Lines 157-158: “The low CT values observed also suggest a low level of

parasitemia...”  Shouldn’t it be “ The high CT values…”?  

>It should, this has been correction in the manuscript.

Line 178:  “…to extract total DNA using the QIAamp RNeasy kit…” Is this correct?

>This has been corrected to Dneasy kit.

Reviewer 2 Report

Comments for Johnson et al., 2021

This well written manuscript by Johnson et. al., 2021 represents a further investigation into the presence of Babesiaspecies pathogens infecting deer in the UK, and these pathogens have been shown to be highly diverse in deer populations across Europe. As alluded to by the authors, deer serve as an important reservoir host for persistence of tick-borne pathogen infections in economically important domestic animals like cattle and sheep. The study analysed infections in deer herds from three different locations in the UK and used a combination of molecular and serological assays to assess for the presence of Anaplasma phagocytophilum Babesia spp., as well as West Nile and Tick-borne encephalitis viruses in samples. DNA sequence analysis was also performed to show the phylogenetic relationships between the Babesia species amplified from samples in this study, with others previously identified in Europe. This study highlights the need of continued monitoring of deer populations for infections with important tick-borne pathogens as these remain possible a threat to not only economically important animal species, but to humans as well. The reviewer therefore suggests the following changes for the improvement of the paper:

Line 47: Gram-negative

Line 47: ‘Anaplasmataceae’ needs to be italicised.

Line 95: When the authors say, ‘Piroplasm was detected’ do they mean that they used microscopic analysis of thin blood smears to observe piroplasms? This statement certainly alludes to this and may be misleading to the reader. If only molecular analysis was used to detect piroplasm DNA, then this should be stated clearly here.

Line 97: Did you mean Dual rather than ‘Duel’ infections?

Figure 1A: The logic of performing this alignment with these particular sequences is not abundantly clear to the reviewer. This needs further clarification in the methods section. Also, why were the Babesia spp. sequences generated in this paper not included in the alignment and identified as such?

Figure 1B: The Babesia species pathogen sequences found in this study need to be highlighted for ease of locating them in the phylogenetic tree. Critically, in your tree construction, did you perform best-fit model testing in MEGA to evaluate which model would give the best phylogenetic Neighbour Joining tree with your data?

Lines 143 – 144: Further molecular characterisation of the A. phagocytopilum pathogens detected in the five animals in this study seems to have been neglected. It would be interesting if data on how these relate to other strains in the UK.

Lines 159 – 160: The reviewer believes the latter may be true as the low Ct may be due to the low DNA quality and quantity of DNA extracted from this sample. Future efforts to perform similar studies should aim to collect EDTA whole blood, which is a well-documented source of DNA for performing molecular assays such as real-time PCR with reliable results. If there are initial sample volume limitations, collections of 1 ml EDTA aliquots prior to separation for serum collection, would be more than adequate. Also, besides WNV and TBEV, were the samples also confirmed to be Anaplasma and Babesia positive using ELISA?

Line 193: Publicly

Based on your Table 1 and 1 data, how do the molecular and serology-based results for presence of A. phagocytophilum and Babesia, as well as WNV and TBEV compare to those in previous studies in the UK and Europe? This could be given more emphasis in the discussion.

Author Response

This well written manuscript by Johnson et. al., 2021 represents a further investigation into the presence of Babesiaspecies pathogens infecting deer in the UK, and these pathogens have been shown to be highly diverse in deer populations across Europe. As alluded to by the authors, deer serve as an important reservoir host for persistence of tick-borne pathogen infections in economically important domestic animals like cattle and sheep. The study analysed infections in deer herds from three different locations in the UK and used a combination of molecular and serological assays to assess for the presence of Anaplasma phagocytophilum Babesia spp., as well as West Nile and Tick-borne encephalitis viruses in samples. DNA sequence analysis was also performed to show the phylogenetic relationships between the Babesia species amplified from samples in this study, with others previously identified in Europe. This study highlights the need of continued monitoring of deer populations for infections with important tick-borne pathogens as these remain possible a threat to not only economically important animal species, but to humans as well. The reviewer therefore suggests the following changes for the improvement of the paper:

Line 47: Gram-negative?

> Corrected

Line 47: ‘Anaplasmataceae’ needs to be italicised.

> Corrected

Line 95: When the authors say, ‘Piroplasm was detected’ do they mean that they used microscopic analysis of thin blood smears to observe piroplasms? This statement certainly alludes to this and may be misleading to the reader. If only molecular analysis was used to detect piroplasm DNA, then this should be stated clearly here.

> The statement has been corrected to “piroplasm DNA”.

Line 97: Did you mean Dual rather than ‘Duel’ infections?

> Corrected to dual.

Figure 1A: The logic of performing this alignment with these particular sequences is not abundantly clear to the reviewer. This needs further clarification in the methods section. Also, why were the Babesia spp. sequences generated in this paper not included in the alignment and identified as such?

> The purpose of the alignment was to highlight a single base-pair deletion in the deer-associated Babesia spp, which could be used for discriminating the different ruminant species. The sequences from this study were not included because they were identical to the sequence MG344775.The text has been revised to state that the sequences were identical.

Figure 1B: The Babesia species pathogen sequences found in this study need to be highlighted for ease of locating them in the phylogenetic tree. Critically, in your tree construction, did you perform best-fit model testing in MEGA to evaluate which model would give the best phylogenetic Neighbour Joining tree with your data?

> Arrows have been inserted to indicate sequences derived from this study. For tree selection we used the tree that illustrated associated with deer-associated Babesia spp. and the separation from the other species detected in the United Kingdom.

Lines 143 – 144: Further molecular characterisation of the A. phagocytopilum pathogens detected in the five animals in this study seems to have been neglected. It would be interesting if data on how these relate to other strains in the UK.

> This study is limited to the detection of DNA suggesting infection. Further analysis of these using different fragments of DNA are being used for this purpose but are the subject of a further publication.

Lines 159 – 160: The reviewer believes the latter may be true as the low Ct may be due to the low DNA quality and quantity of DNA extracted from this sample. Future efforts to perform similar studies should aim to collect EDTA whole blood, which is a well-documented source of DNA for performing molecular assays such as real-time PCR with reliable results. If there are initial sample volume limitations, collections of 1 ml EDTA aliquots prior to separation for serum collection, would be more than adequate. Also, besides WNV and TBEV, were the samples also confirmed to be Anaplasma and Babesia positive using ELISA?

> We agree with the reviewers comments that EDTA blood would be a more appropriate sample and should be used in future studies. We did not attempt to confirm serological confirmation of the molecular findgings using ELISA but if these assays were available it would provide another approach to this type of investigation.

Line 193: Publicly

> Corrected

Based on your Table 1 and 1 data, how do the molecular and serology-based results for presence of A. phagocytophilum and Babesia, as well as WNV and TBEV compare to those in previous studies in the UK and Europe? This could be given more emphasis in the discussion.

> We do highlight that the findings from this study corroborate the previous studies from the UK (reference 35). However, due to the issues of cross-reactivity and the relatively small study presented, we are wary of overstating our findings compared to others (reference 37).

Reviewer 3 Report

In your study, you have detected 12% of sera positive for tick-borne Encephalitis virus. However, you didn't provide any data of virus detection from the ticks. It would be nice to get a supporting data to validate your report.

Additionally, there is an another report about tick-borne encephalitis virus from UK (Holding et al 2020, EID). How is your study different from theirs and contribute to new information?

Similar to the earlier comment, it would be nice to add protozoan detection data from tick.

Minor corrections:

Line 97: Duel? Dual?

Line 97: which are the pathogens?

Line 177: haeolysed? haemolysed?

Line 178: RNeasy kit? should it be DNAeasy kit?

Author Response

In your study, you have detected 12% of sera positive for tick-borne Encephalitis virus. However, you didn't provide any data of virus detection from the ticks. It would be nice to get a supporting data to validate your report.

> We agree that matched data from ticks collected either from the culled animals or from the site they grazed would be of value but beyond the scope of this study.

Additionally, there is an another report about tick-borne encephalitis virus from UK (Holding et al 2020, EID). How is your study different from theirs and contribute to new information?

> We site the Holding et al study (reference 35), acknowledge their findings and highlight that our TBEV data corroborates their finding. However, the main focus of this study was detection of piroplasm and Anaplasma phagocytophilum DNA within captive deer samples, something that the previous study did not investigate.

Similar to the earlier comment, it would be nice to add protozoan detection data from tick.

> See comment above.

Minor corrections:

Line 97: Duel? Dual?

> Corrected to Dual.

Line 97: which are the pathogens?

> Corrected to “piroplasm and A. phagocytophilum”

Line 177: haeolysed? haemolysed?

> Corrected to "haemolysed".

Line 178: RNeasy kit? should it be DNAeasy kit?

> Corrected to "DNeasy kit"

Round 2

Reviewer 3 Report

The authors have addressed the comments and I don't have any major issues with this manuscript.

Minor comment:

Line 69: Does Babesia need to be underlined?